# Construction of a reference transcriptome for the analysis of male sterility in sugi (*Cryptomeria japonica* D. Don) focusing on *MALE STERILITY 1* (*MS1*)

Fu-Jin Wei[1¤], Saneyoshi Ueno[1]*, Tokuko Ujino-Ihara[1], Maki Saito[2], Yoshihiko Tsumura[1,3], Yuumi Higuchi[4], Satoko Hirayama[5], Junji Iwai[4], Tetsuji Hakamata[6], Yoshinari Moriguchi[7]

1 Forestry and Forest Products Research Institute, Forest Research and Management Organization, Ibaraki, Japan, 2 Forest Research Institute, Toyama Prefectural Agricultural Forestry and Fisheries Research Center, Toyama, Japan, 3 Faculty of Life and Environmental Sciences, University of Tsukuba, Ibaraki, Japan, 4 Niigata Prefectural Forest Research Institute, Niigata, Japan, 5 Niigata Regional Promotion Bureau, Niigata, Japan, 6 Shizuoka Prefectural Research Institute of Agriculture and Forestry, Shizuoka, Japan, 7 Graduate School of Science and Technology, Niigata University, Niigata, Japan

¤ Current address: Institute of Plant and Microbial Biology, Academia Sinica, Taipei, Taiwan
* saueno@ffpri.affrc.go.jp

**Data Availability Statement:** Sequences and assembled transcripts presented in the current study have been deposited at DDBJ/EMBL/

## Abstract

Sugi (*Cryptomeria japonica* D. Don) is an important conifer used for afforestation in Japan. As the genome of this species is 11 Gbps, it is too large to assemble within a short time-frame. Transcriptomics is one approach that can address this deficiency. Here we designed a workflow consisting of three stages to *de novo* assemble transcriptome using Oases and Trinity. The three transcriptomic stage used were independent assembly, automatic and semi-manual integration, and refinement by filtering out potential contamination. We identified a set of 49,795 cDNA and an equal number of translated proteins. According to the benchmark set by BUSCO, 87.01% of cDNAs identified were complete genes, and 78.47% were complete and single-copy genes. Compared to other full-length cDNA resources collected by Sanger and PacBio sequencers, the extent of the coverage in our dataset was the highest, indicating that these data can be safely used for further studies. When two tissue-specific libraries were compared, there were significant expression differences between male strobili and leaf and bark sets. Moreover, subtle expression difference between male-fertile and sterile libraries were detected. Orthologous genes from other model plants and conifer species were identified. We demonstrated that our transcriptome assembly output (CJ3006NRE) can serve as a reference transcriptome for future functional genomics and evolutionary biology studies.

GenBank with accession numbers DRR174638-DRR174656 and ICQT01000001-ICQT01048643, respectively. The ForestGEN database also hosts the cDNA and translated protein sequences (https://forestgen.ffpri.go.jp/CJ3006NRE/clusterList), which are freely accessible. Scripts used in this analysis are available at Bitbucket (https://bitbucket.org/saueno1/masker/src/master/) and Figshare (https://doi.org/10.6084/m9.figshare.13726528.v1).

**Funding:** Grant-in-Aid from the Science and technology research promotion program for agriculture, forestry, fisheries and food industry (No.28013B) (YM) Funder name: Ministry of Agriculture, Forestry and Fisheries of Japan(MAFF) and NARO Bio-oriented Technology Research Advancement Institution(BRAIN) http://www.naro.affrc.go.jp/laboratory/brain/innovation/ The funders had no role in study design, data collection and analysis, decision to publish, or preparation of the manuscript. JSPS KAKENHI (Grant Number 25450223)(SU) Funder name: Japan Society for the Promotion of Science https://www.jsps.go.jp/english/index.html The funders had no role in study design, data collection and analysis, decision to publish, or preparation of the manuscript. Research grants #201119 (SU) Funder name: Forestry and Forest Products Research Institute https://www.ffpri.affrc.go.jp/ The funders had no role in study design, data collection and analysis, decision to publish, or preparation of the manuscript. Research grants #201421 (AM) Funder name: Forestry and Forest Products Research Institute https://www.ffpri.affrc.go.jp/ The funders had no role in study design, data collection and analysis, decision to publish, or preparation of the manuscript. Research program on development of innovative technology (No.28013BC) (YM) Funder name: NARO Bio-oriented Technology Research Advancement Institution http://www.naro.affrc.go.jp/laboratory/brain/innovation/ The funders had no role in study design, data collection and analysis, decision to publish, or preparation of the manuscript.

**Competing interests:** The authors have declared that no competing interests exist.

## Introduction

*Cryptomeria japonica* D. Don (Japanese cedar), also known as "sugi" in Japan, is a large evergreen conifer tree species. Due to its swift growth and its adaptation to most environments in Japan, it represents an important species for the forestry industry. After World War II, sugi plantations have increased to comprise 42% of Japan's artificial forests [1]. Therefore, the demand for elite tree varieties is a primary reason to obtain more knowledge of sugi genomics; however, other motivations can include medical and economic reasons. Sugi pollen leads to severe allergy in about 25% of the Japanese population [2]. Planting male-sterile sugi is one possible solution to this problem [3], as at least 23 male-sterile trees [4], each with one of the four independent recessive alleles (*ms1*, *ms2*, *ms3*, and *ms4*) responsible for male sterility, has been discovered and could be used for breeding [5]. Recent advances in technology have revealed more details of these loci by a functional genomics study via transcriptomics approach to more precisely pinpoint the genetic variation or genes related to male sterility [6, 7], as a reference genome sequence of sugi is not currently available.

It is difficult to study functional genomics in gymnosperms because of their long life span and the large genome size. For instance, the genome sizes of Norway spruce (*Picea abies*) [8], white spruce (*Picea glauca*) [9, 10], loblolly pine (*Pinus taeda*) [11], ginkgo (*Gingko bioloba*) [12], and sugi (*Cryptomeria japonica*) [13] are 20 Gb, 20.8 Gb, 20.15 Gb, 11.75 Gb, and 11 Gb, respectively. So, far, only seven gymnosperm genomes have been published—the Norway spruce [14], white spruce [9, 10], loblolly pine [11], sugar pine [15], Douglas fir [16], ginkgo [12] and gnetum [17]. Decoding the genome using appropriate annotations is essential for determining the functions of individual genes. There has recently been an increase in the use of the emerging annotation softwares "MAKER" and "MAKER-P" for plant species [18]. Genomic information from assembled genome sequences, RNA, and protein data can be combined and annotated. Abundant RNA or protein data is required for the annotation process; therefore, even for species for which the complete genome sequence is available, transcriptome analysis is still needed for a functional genomic study [8, 19–23].

Many assembled expressed sequence tags (EST) for sugi had been published in public databases [24, 25] before the availability of high throughput sequencing technology, such as next generation sequencing (NGS). However, NGS has proven to be greatly beneficial for the advancement of functional genomic research, due to its increased yields, reduced unit price, and multiple analysis tools available [7, 26]. Choosing a suitable assembly tool [27] and integration of the transcriptome is a good practice, which has been performed in *Abies sachalinensis*, another conifer species [28].

Precise sequence information of the transcriptome is an essential step for future work; here we aimed to construct a high-quality cDNA assembly of sugi. We independently assembled transcriptome data from 10 different genome accessions of sugi. These were then integrated via a semi-manual process and merged with outputs of the EvidentialGene software [29]. To find and identify the male strobili-specific sterility genes, 10 RNA-Seq libraries of uneven runs were scrutinized. By referencing the benchmark testing and coverage of and by different cDNA sources, our assembled cDNA was evaluated. Using the reference transcriptome constructed in this study, we were able to analyze gene expression, and identify variants and orthologs, focusing on *MALE STERILITY 1* (*MS1*) in sugi. This transcriptome resource will be useful for future sugi breeding and genetic research.

## Materials and methods

### Plant materials

Ten accessions were prepared for mapping the male-sterile genes. The pedigree is shown in S1 Fig. 'Nakakubiki-4' is the male parent of 'T1NK4F1.' T5_normalMIX_ms1 and T5_sterile-MIX_ms1 are from the progeny of 'T1NK4F1' backcrossed with 'Toyama MS,' and represent the male-fertile *Ms1/ms1* and male-sterile *ms1/ms1*, respectively. These four accessions are hereafter referred to as the T5 family. S3T67_normalMIX_ms1 and S3T67_sterileMIX_ms1 represent samples from the progeny of 'T1NK4F1' (male-fertile) backcrossed, in terms of *MS1*, with 'Shindai-3' (male-sterile). 'Ooi-7' is heterozygous at the male-sterile gene (*Ms1/ms1*). 'S1NK4' is the F1 hybrid of 'Shindai-1' and 'Nakakubiki-4.' 'S5HK7' is the F1 hybrid of 'Shindai-5' and 'Higashikanbara-7,' and 'S8HK5' is the F1 hybrid of 'Shindai-8' and 'Higashikanbara-5.' Most parental lines of these crosses above are not included in this study excepting only 'Nakakubiki-4,' however, these lines provided four unique male-sterile genes. The genetic characters of these male-sterile genes have been well studied, and they are all recognized to be recessive genes. In these samples, only T5_sterileMIX_ms1 and S3T67_sterileMIX_ms1 contain male-sterile groups, because these samples consisted only of male bud tissue.

### Extraction of RNA and sequencing

Ten RNA-Seq libraries were constructed from 10 different accessions. For S3s (T5_normal-MIX), S4s (T5_sterileMIX), S5s (S3T67_normalMIX), and S6s (S3T67_sterileMIX), RNA was extracted from several (up to 50) individuals and the mixture of RNA from progeny in each sample was sequenced. RNA-Seq was performed for S1s to S6s, as described in [22]. For the remaining libraries, RNA was extracted from different tissues (Table 1) following the method used in [22]; the mixture of RNA from these tissues was sequenced on a HiSeq2000 platform (Illumina) by Hokkaido System Science Co., Ltd, Sapporo, Japan. We performed ISO-Seq (isoform sequencing) using the RNA of S2s and sequenced on PacBio RS II (Pacific Biosciences) with four cells by P3C5 chemistry by Takara Bio Inc., Shiga, Japan.

**Table 1. Statistics of raw and mapped reads.**

| Working ID | Accession | Tissue[&] | Seq. Tech.[$] | Read pairs | Contig # | Gene # | Max. length |
|---|---|---|---|---|---|---|---|
| **S1s** | Nakakubiki-4 | MF | H&M | 79,807,618 | 100,433 | 34,937 | 16,581 |
| **S2s** | T1NK4F1 | MF | H&M | 80,381,867 | 98,141 | 33,527 | 16,394 |
| **S3s** | T5_normalMIX | MF | H&M | 227,248,962 | 124,448 | 42,861 | 17,669 |
| **S4s** | T5_sterileMIX | MF | H&M | 231,666,349 | 119,627 | 41,875 | 20,848 |
| **S5s** | S3T67_normalMIX | MF | H&M | 5,414,221 | 43,326 | 22,531 | 12,984 |
| **S6s** | S3T67_sterileMIX | MF | H&M | 55,126,058 | 97,360 | 36,363 | 16,997 |
| **Ooi-7** | Ooi-7 | IBL | H | 60,826,308 | 82,064 | 30,799 | 16,753 |
| **S1NK4** | Shindai-1 × Nakabuki-4 | MF&IBL | H | 53,857,318 | 94,578 | 34,252 | 16,377 |
| **S5HK7** | Shindai-5 × Higashikanbara-7 | MF&IBL | H | 55,303,023 | 88,771 | 32,348 | 17,018 |
| **S8HK5** | Shindai-8 × Higashikanbara-5 | IBL | H | 50,816,212 | 81,387 | 31,021 | 16,221 |
| **CJ3006All** | Integrated | MF&IBL | H&M | - | 116,466 | - | 20,997 |
| **CJ3006NRE** | Integrated | MF&IBL | H&M | - | 49,758 | - | 17,669 |

[&]: MF: Male flower; IBL: Inner bark & leaf.

[$]: H&M: HiSeq and MiSeq; H: HiSeq.

The minimum length of each library is 501 bps.

## Assembly and annotation

**Quality control and independent assembly.** Before assembly, the raw reads were passed through four quality control filters using cutadapt ver. 1.5 [30]. These included: (1) cutting 13 bases from the 5'-ends, (2) cutting over-reading due to adapters or primers, (3) cutting low quality base tails, and (4) setting the minimum length threshold to 35 bases after these other steps. The number of filtered reads are listed in Table 1. Hereafter, "library" is used to indicate one or multiple runs of RNA-Seq data from the same accession or variety.

All 10 libraries were assembled using two different softwares—Oases v0.2.08 [31] and Trinity v 2.4.0 [32]. Since the maximum k-mer for Trinity was 32, we performed only a single run for each library. For Oases (Velvet), we used the k-mer in the range from 35 to 43 (odd numbers), with five runs for each library. For both softwares programs, the minimum length of contigs was 500 and 300 for Trinity and Oases, respectively.

**Integration.** Contig integration was used to overcome the sampling bias. We used two integration methods to increase the reliability of integrated contigs (S2 Fig). One method was an automatic pipeline using EvidentialGene [29], which is an open source software. The other was a semi-manual pipeline using homemade scripts to manipulate the assembled result from Trinity library assembly. The coding languages used include shell script, AWK, and Python.

In the workflow through EvidentialGene [29], each library was processed by tr2aacds.pl ver. "2016.07.11" (downloaded from http://arthropods.eugenes.org/EvidentialGene/other/evigene_old/evigene_older/) to produce 10 independently integrated contigs (FASTA files). This step was intended to retain the isoforms that were produced under different conditions. Then, the 10 contig FASTA files were concatenated into one, and again processed by tr2aacds.pl (second integration). The only customized parameter was setting the minimum length of the CDS to 90, i.e. "—MINCDS90". After the second integration using EvidentialGene, there were 107,674 transcripts which could be converted into 108,886 protein sequences.

At the beginning of the semi-manual integration, the assembled contigs have been translated to the longest open reading frame as the representative protein sequences. The semi-manual integration process included three modules: 1) finding the seed sequence, 2) mining more paralogs, and 3) abstracting by homologs; named module-1, module-2, and module-3, respectively. Both nucleotide and protein sequences were used for alignments, depending on whether the alignments were within species or among species. Scripts has been uploaded to Bitbucket repository.

In module-1, the main objective was to select seed sequences via BLASTX. We expected these seeds contained the functional domain. These do not necessarily have to be orthologs of the database sequence (subject). We used SwissProt [33] (downloaded June 11, 2017) as a reference database to find the orthologous genes for each library. Two filters were applied to the BLASTX result: 1) Minimum ratio of the sum of HSP against the subject's sequence was 50%; 2) Minimum HSP length was 20 amino acids. The contigs with the best score were selected for every SwissProt gene. Thus, one SwissProt gene was only linked to one contig. This BLASTX search was performed reciprocally and we identified orthologous sequences between each library and SwissProt genes based on reciprocal best blast hit. After this step, we obtained one set of cDNA sequence used as the "seed" sequence for the next step.

In module-2, we aimed to extend the previous step and concentrate to identify paralogs. Similar genes have been identified in module-1 as seed genes, which represented the most similar genes between angiosperms and the gymnosperm (*C. japonica*). Any duplication event after the divergence of angiosperms and gymnosperms, which produced the paralogs, could be identified from the seed genes in this step. The "extend" query used these seed sequences to identify more paralogs. We used seed sequences as a query against all 10 libraries, including

the source library of the seed. There were no customizations in this BLAST step. Since the contigs were from the same species, we expected to retrieve as many homologous contigs in the sequence as possible. The "concentrate" filter (findCommon.py) was used to reduce duplication sequences to be as representative as possible. To find the representative sequence among homologs, the maximum tolerance for mismatches and gaps was 2 bps. In addition, only representative sequence that existed in at least two libraries were retained.

At the last step in module-2, any removed singleton sequences were identified, defined as not only "seed sequences," but also those without homologs in any other library. These singletons were added back before the next step. In module-3, by comparison with the taxonomic information in the NCBI-NR database, we discarded those sequences that matched non-eukaryotes, but kept those that matched eukaryotes and others that were unclassified. Products from both the semi-manual and automatic EvidentialGene pipelines were merged based on BLAST homology into CJ3006ALL, which was then BLASTed again against the NCBI-NR database. Sequences matching eukaryotes were remained as "CJ3006NRE," which is the name of integrated library used for downstream analyses.

**Annotation.** We used two tools, EvidentialGene [29], and InterProScan (v 5.30) [34], to annotate the integrated library (CJ3006NRE). The reference databases for running namegenes. pl (which was included in http://arthropods.eugenes.org/EvidentialGene/other/evigene_old/ evigene_older/evigene17mar10.tar, the annotation tool in EvidentialGene), were UniRef50 (downloaded at May 2018) [35] and CDD (Conserved Domains Database, v 3.16). The reference database used for annotation was InterPro5 v 69.0. Isoforms were identified by BLASTing all contigs with the parameter "word size = 100 bps." Then, contigs were matched to other contigs with over 90% of genes identified. An HSP length over 150 bps was considered an isoform. To identify transcription factors in sugi, we re-scanned all contigs using pfamScan [36]. The list of transcription factors was based on a joint list with the work published in [37] and a Pfam list published online at www.transcriptionfactor.org [38]. To predict metabolic enzymes, we used a pipeline E2P2, downloaded from the "Plant Metabolic Network" (PMN) [39]. In order to mark the repetitive sequences, RepeatMarker v 4.0.7 (http://www.repeatmasker.org) [40] and RepBase v 22.05 were used as reference databases. Orthologous genes were identified in *Arabidopsis thaliana* (ath), *Oryza sativa* (osa), *Picea abies* (pab), and *Pinus taeda* (pta) proteome sequences downloaded from PLAZA web site [41] using the BLASTP program by the orthologr R package [42], which performs a best reciprocal hit (BRH) search between two genomes. The BRH algorithm with BLASTN was also applied to identify probe and contig sequences of *C. japonica* transcripts from other studies [6, 7, 43]

## Evaluation

To evaluate the proportion of input reads that were filtered out during integration, we aligned the input reads to the originally assembled contigs and the integrated contigs, using BWA [44, 45] as a mapping tool. Considering that *Cryptomeria japonica* used in the current study is a heterogeneous species and not an inbred line, we limited the parameter of "penalty." For BWA, we used the "bwa mem" module and one of the parameters set to "-O 4,4," representing penalties for deletions and insertions, which indicated gaps in the reads and in the references, respectively. We also used other RNA-Seq libraries for *C. japonica*, downloaded from NCBI-SRA (S1 Table) and mapped the reads onto CJ3006NRE using BWA with the same parameter settings.

To estimate the coverage of the core orthologous genes, we used BUSCO (v 3.0.2) [46] to test all 10 libraries as well as the integrated library. The testing model was "transcriptome", and the version of the reference database was "embrophyta_odb9". For each of 10 RNA-Seq

libraries, we used their own assembled contigs as the input; for the integrated library, we used CJ3006NRE as the input.

ISO-Seq data were processed using the "pbtranscript-tofu" analysis suite v 1.0.0.177900 [47]. The only customized parameter was the minimum length set to 300 bps. After this process, there were 56,399 transcripts (ISOSeq0215) clustered from three ISO-Seq runs. Within these transcripts, 9,352 transcripts were classified as a high-quality subset, called "ISOSeq0215hq."

A total of 23,111 full-length cDNA sequences (CJ_FLcDNA) were downloaded from NCBI. We retrieved these sequences using the keywords, "Cryptomeria japonica," "full-length," and "cDNA" in the NCBI web-based searching interface on September 13, 2018.

### Differential expression

Using our integrated library CJ3006NRE as a reference, we compared the expression levels among the 10 libraries. When multiple sequencing runs were included in each library, we considered these runs as repetitions and labeled them as "_rep1," "_rep2," and so on (S2 Table). "Kallisto," a package which calculates the building index of the reference sequence and quantifies the abundance from FASTQ files [48], was used to quantify the transcript abundances using bootstrap estimation from 100 repetitions. The output of "Kallisto" was processed in differential expression analysis using the package "sleuth [49]." Correlation of gene expression between RNA-Seq libraries was analyzed with Pearson's product-moment correlation coefficient (PCC) using the R package rsgcc [50]. The expression difference between male-sterile (S3s) and fertile (S4s) libraries was analyzed through time-series samples (from _rep1 to _rep5 in S2 Table).

### Gene Ontology (GO) annotation

GO terms were assigned with InterProScan during the annotation process. The classification of the GO terms was performed with CateGOrizer [51] using Plant_GOslim as the classification list.

### Variant calling

Upon obtaining the mapped files in BAM (binary SAM) format, we used samtools [52] and bcftools (https://github.com/samtools/BCFtools) to call the variants with a customized filter of QUAL > 20 and DP > 3 (variant quality value greater than 20 and sequence depth of the variant site greater than 3). Group-specific variants were classified using the "isec" command in bcftools. Appendix S1 shows an example of command lines. According to the pedigree (S1 Fig), we extracted the group (or accession)-specific variants. E.g., for 'Shindai 3,' there was only one site with allele "1" in S5s and S6s, and allele "1" was not present in the rest of the sites (S3 Fig). Relatedness ($r_{xy}$ in Hedrick et al. [53]) was calculated among accessions using ngsRelateV2 [54], after a merged vcf file was filtered to prepare a complete data set with vcftools [55].

### Results

The transcriptome assembly in the present study resulted in 49,758 transcripts (CJ3006NRE), which were constructed using a series of manipulations to 10 individual RNA-Seq libraries. We used two RNA-Seq *de novo* assembling tools (Oases [31] and Trinity [32]), and applied two different methods to merge the transcriptomic sequences with the RNA-Seq libraries (automatic assembly using EvidentialGene [29] and semi-manual assembly in multiple steps).

The results (Fig 1A) showed the highest effect by tissue tested (Table 1) compared to the pedigree of samples (S1 Fig). A heatmap of gene expression (Fig 1B) clearly showed that RNA samples were clustered according to tissue type. Male strobili samples (S1s to S6s) clustered together, and deviated from the inner bark and leaf samples (Ooi-7 and S8HK5).

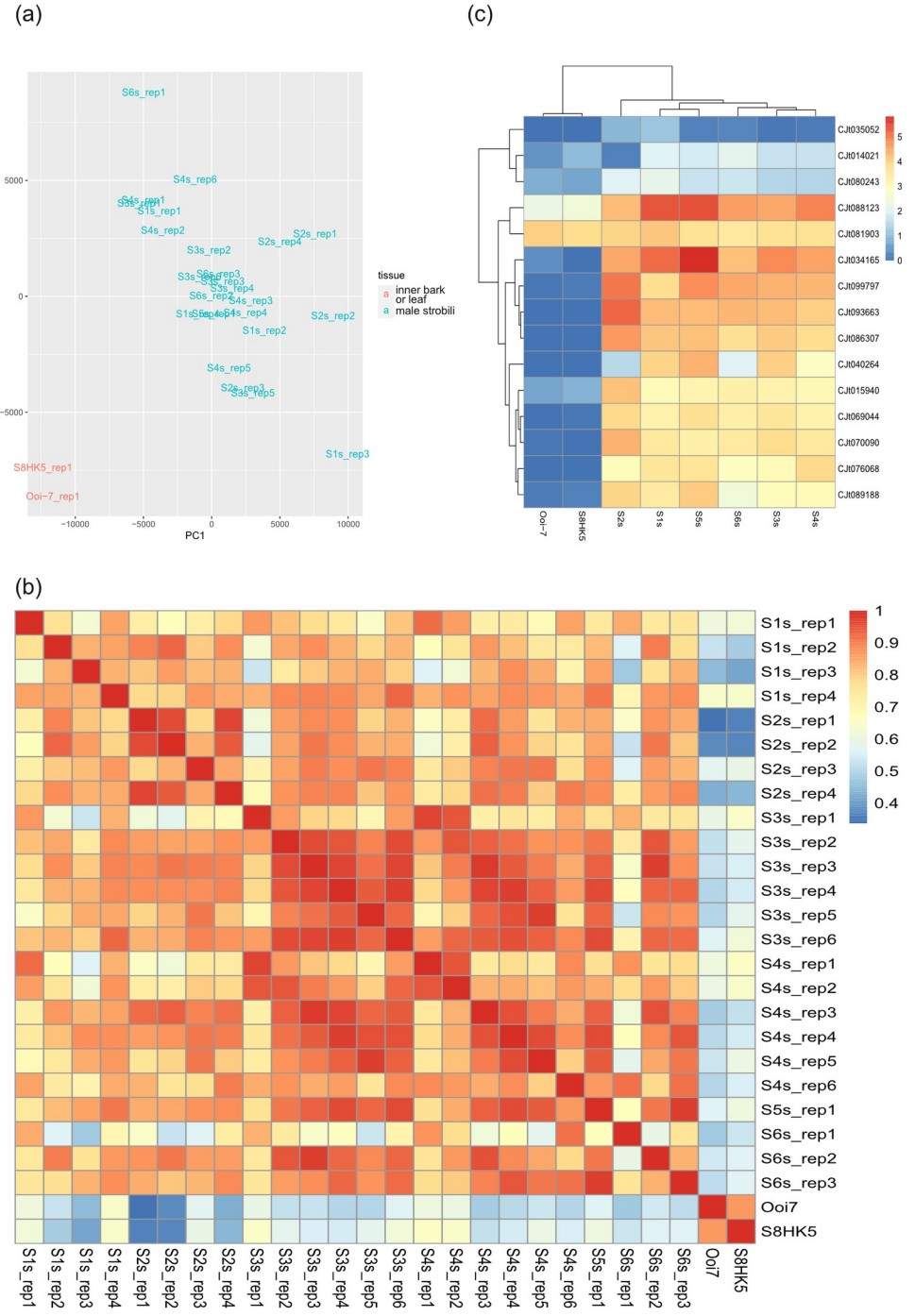

**Fig 1. Assessment of the gene expression data.** (a) The principal component analysis (PCA) results of each run. Green color indicates samples collected from male strobili. Red color indicates samples from the inner bark or leaf tissue. (b) Paired similarity heatmap among each run. Darker red and blue colors indicate higher similarity and divergence, respectively. (c) Heatmap of gene expression of potential genes downstream of MYB80s.

## Assembly and annotation of the EST library

There were 116,466 and 49,758 contig sequences in CJ3006All (the contig set without any filtering) and CJ3006NRE (only CJ3006All contigs that matched to eukaryote in NCBI-NR), respectively. Using Trinity, the number of contigs per library ranged from 43,326 (S5s) to 124,448 (S3s), while in Oases, the number of contigs per library ranged from 34,303 (S5s) to 105,184 (S3s) (S3 Table). The number of contigs was affected by the number of reads per library (Table 1).

There were a total of 31,678 and 47,968 genes among the 49,758 contigs to which we assigned a functional annotation using InterProScan [34] and EvidentialGene [29], respectively. It was difficult to identify real gene isoforms from the assembled transcriptomes without genomic sequences; in this study, we used a self-BLAST search to identify isoforms, and found 17,079 gene isoforms (S4 Table) within CJ3006NRE (49,758 transcripts). Although these isoforms could be used as representatives, we elected to use all 49,758 transcripts as individual genes in subsequent analyses. In total, 1,291 genes related to transcription factors were identified by Pfam [36] (S4 Table). Among these, 974 genes were considered to be unique, without isoforms (S4 Table).

Using RepeatMasker [40], the numbers of transposable elements within CJ3006NRE were estimated to be 7,029 and 2,282 for retrotransposons (Class 1) and DNA transposons (Class 2), respectively. Repetitive sequences made up approximately 4.1% of all cDNA sequences in CJ3006NRE (S5 Table). The majority of this repetition was LTR (long terminal repeat) elements, forming about 2.54% of nucleotide bases in the total length of CJ3006NRE. However, only 3,105 and 754 elements were identified and classified as class I and class II transposable elements following the customized filtering of the coverage rate ($> 20\%$) and the length of HSP ($> 200$ bps) (S5 Table). Although retrotransposons may account for the large size of the sugi genome, most of these may be silenced in the 10 RNA-Seq libraries collected. The metadata and annotation are presented in a Supplementary Data (S4 Table).

Orthologous sequences were identified for 9,373 (34.2%) *A. thaliana*, 9,074 (22.3%) *O. sativa*, 11,163 (16.8%) *P. abies*, and 9,910 (11.7%) *P. taeda* proteome sequences. Similarly, probe and contig sequences of *C. japonica* transcripts from previous studies were identified on CJ3006NRE (S4 Table).

## The benchmark of assembly

We evaluated our assembled library with the benchmark tool (BUSCO version 3.0.2 [46]) using the built-up core reference database (embrophyta_odb9), and we compared our integrated EST library to other cDNA resources (Fig 2). When the ratio of missing parts of CJ3006NRE to the reference database was compared with that of other assembled libraries (S1s to S8HK5), it was found to be lower in CJ3006NRE (10.76%). Furthermore, completeness, especially the "Complete (C) and single-copy (S)" category, was clearly higher (78.47%) than other libraries (41.88% - 49.24%). In general, the benchmark of CJ3006NRE is higher than any other single assembled library.

## The coverage of different full-length cDNA libraries

In order to evaluate the advantage of the RNA-Seq technology and integration method used in this study, we compared the coverage between CJ3006NRE and three other cDNA sources, full-length cDNA by Sanger sequencing (CJ_FLcDNA) and full-length cDNA by ISO-Seq in our own laboratory (ISOSeq0215 and ISOSeq0215hq).

The full-length cDNA library was retrieved using the keywords "Cryptomeria japonica," "cDNA," and "full-length" in the NCBI-Nucleotide database on September 13, 2018. In total,

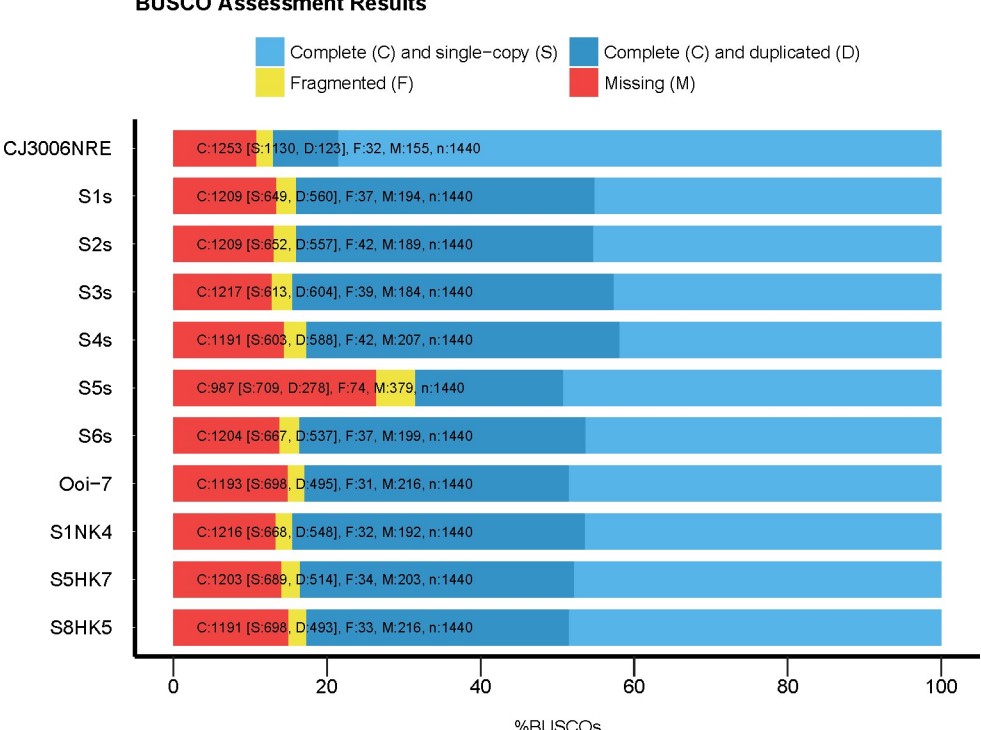

**Fig 2. The benchmark of the assembled contigs.** The y-axis shows the samples listed in Table 1, and the x-axis (% BUSCO) indicates the percentage of assembled contigs for each category (Complete and single-copy, Complete and duplicated, Fragmented, and Missing). The reference database for BUSCO (v 3.0.2) was "embrophyta_odb9" with 1,440 core genes in total.

23,111 nucleotide sequences were retrieved, downloaded, and formatted as a BLAST database, called "CJ_FLcDNA."

To screen on the BLAST results, we merged the High-scoring Segment Pairs (HSP) of each query-subject pair. Then, we separately calculated the coverage of the query and the subject. In Table 2, we only counted sequences where the coverage was over 75% of the total length. The transcript coverage of each cDNA source by the library CJ3006NRE ("Covered by CJ3006NRE") was higher than the coverage of CJ3006NRE by other cDNA resources ("Cover to CJ3006NRE"); 10,787 (19.13%) and 7,208 (31.19%) sequences in ISOSeq0215 and CJ_FLcDNA were not covered by CJ3006NRE, respectively.

## The mapping rate of RNA-Seq reads

The mapping rate of the reads was calculated based on the statistics of the mapping file (BAM), by dividing the number of unmapped reads by the total number of reads of each

**Table 2. The cDNA source and coverage by and to CJ3006NRE.**

| cDNA source | Covered by CJ3006NRE† | Cover to CJ3006NRE‡ |
|---|---|---|
| ISOSeq0215 | 45,612 (80.87%) | 21,498 (43.21%) |
| ISOSeq0215hq | 8,826 (94.38%) | 9,178 (18.45%) |
| CJ_FLcDNA | 15,903 (68.81%) | 16,814 (33.79%) |

† The percentage is a ratio of CJ3006NRE coverage to coverage by each cDNA source (56,399, 9,352 and 23,111 for ISOSeq0215, ISOSeq0215hq and CJ_FLcDNA, respectively) with blast hits.

‡ The percentage is a ratio of coverage of each cDNA resources to CJ3006NRE (49,758) with blast hits.

library, and then subtracting from one. Since the integration process occurred over several steps, we used the mapping rate (ratio of mapped reads to the total number of reads in the sequencing library) to reveal how much information was lost in the sequencing. Using CJ3006NRE as the reference, the mapping rate ranged from 91.99% to 96.49% (S6 Table). Theoretically, the mapping rate against the contigs assembled from the queried reads should be one hundred percent; however, only 96.27% (S5s) to 99.04% (S4s) was achieved. After the integration, however, almost all mapping rates were reduced by approximately 0.87% (S3s) to 2.09% (S1s), except for S5s, which increased by 0.74%. By filtering out non-eukaryote assemblies (i.e., contamination), the mapping rate was reduced by an additional approximately 1.58% (S4s) to 3.35% (S1NK4). Thus, there was a low number of reads that were discarded by the integration and filtering processes. To determine the extent to which reads were enriched by manual integration, the column "Only EviGene" was created (S6 Table). This calculation showed enrichments of 1.09% (S4s) to 2.31% (S1NK4). Compared to the number of manually integrated contigs, 7,580 contigs (15.23% of CJ3006NRE), a modest enrichment was given by the manual integration.

Based on the alignment of CJ3006NRE contigs to full-length cDNA sequences downloaded from NCBI, we estimated how much the RNA-Seq data covered the full-length cDNA library, particularly for the cDNA collected from the same plant organ, the male-strobilus (S3s to S6s library). The results showed a distribution of 67.92% (S6s) to 84.69% (S4s) (S6 Table). The two highest mapping rates were contributed by two libraries with the highest numbers of reads, S3s and S4s. This suggests that the full-length cDNA from NCBI only covers less than 85% of the total RNA, based on the assumption that RNA-Seq represented the total RNA.

When RNA-Seq reads from other studies were mapped against CJ3006NRE, the mapping rate ranged from 93.05% to 98.51% for cambium and male strobili tissue, respectively (S1 Table). This indicates that CJ3006NR is appropriate for the analysis of male sterility.

## Differential expression

The 10 RNA-Seq libraries could be classified into three groups, according to the type of tissue sample collected: 1) all RNA from male strobili; 2) all RNA from the inner bark and leaf (IBL), and 3) mixed with male strobili, inner bark, and leaf materials. Before looking at the differential expression between any certain pair of groups or libraries, principal component analysis (PCA) and sample heatmaps were used to reflect the characteristics of these libraries. The two libraries collected from only IBL tissues (Ooi-7 and S8HK5) showed very different gene expression patterns (expressed genes differed) from the other libraries (Fig 1). In the PCA, either the first or second principal component divided the libraries from bark and leaf from the accessions from male strobili (Fig 1A). Although most of the samples were collected from male strobili, bias during sampling or assembling may have occurred. In Fig 1B, the darker blue color indicates a higher dissimilarity between two accessions, as indicated by the X-axis and Y-axis. With the exception of "Ooi-7 against S8HK5," almost all accessions compared against Ooi-7 or S8HK5 showed high dissimilarity ($> 0.1$, darker color). This supports the previous PCA results in one-to-one comparisons.

Two independent runs of the S1s assembly (i.e., S1s_rep2 and S1s_rep3) showed high dissimilarity to several other runs collected at different times. The reason for the dissimilarity in these two runs from the others is uncertain; however, two other runs of S1s (i.e., S1s_rep1 and S1s_rep4) supported the similarity of S1s to those from the male-strobilus libraries. Nonetheless, the PCA showed that S1s_rep1, S1s_rep2, and S1s_rep4 were clustered with the other male-strobilus accessions. Thus, the divergence did not have a large effect on the overall clustering.

The gene expression in MF (library S1s to S6s) and IBL (library Ooi-7 and S8HK5) samples (Table 1) significantly differed in 7,776 genes (P-value < 0.05) (S7 Table). Among these, 4,471 were upregulated in the male strobili, with a fold change ranging from $2^{0.33}$ to $2^{10.28}$. The range of fold change for the remaining 3,305 downregulated genes was from $2^{-6.41}$ to $2^{-0.33}$.

There was a total of 377 transcription factors within these 7,776 differentially expressed genes (DEGs) (S7 Table). Of these, the three largest gene families were MYB DNA-binding, HLH, and AP; which consisted of 56, 43, and 42 genes, respectively. The number of differentially expressed transcription factors in the male strobili was about 3 times higher than in the leaf or inner bark, indicating that activities occurring in male strobili may require higher gene regulation.

## Variant calling and gene expression

Variations among the accessions compared against CJ3006NRE were summarized in S8 Table, ranging from 26,778 to 49,488 variations observed in 'Nakakubiki-4' and 'Toyama MS,' respectively. Correlation between relatedness ($r_{xy}$) and level of gene expression (PCC) for pairs of accessions was examined by Mantel test using the R package vegan [56]. We found a significant correlation (Spearman's rank correlation $r = 0.7207$, $P = 0.0004$) (S4 Fig), which reflected the experimental design of the current study, wherein male flower samples (MF) were collected from the offspring of the linkage-mapping family, with higher relatedness expected here than in other comparisons. The correlation of gene expression between libraries was generally high when both libraries were constructed from the same tissue type regardless of the relatedness.

## Discussion

The CJ3006NRE dataset is the integrated product assembled from 10 different libraries. We used two different approaches (an automatic EvidentialGene pipeline and a semi-manual series of BLASTs) to integrate each RNA-Seq library into a single assembled library. We then performed a second integration to unite the two integrated results into CJ3006NRE. After eliminating potential contamination by BLASTing against the NCBI-NR database and filtering out potential assembly error, the resulting sequence CJ3006NRE was deemed suitable for investigating differential gene expression and making structural annotations.

### Sequencing depth is the keys to accurate assembly

A suitable method of *de novo* assembly for transcriptome is still under discussion [57, 58]. A high sequencing depth or number of reads is a key factor in the sampling of all transcripts. In relation to the benchmark for S1s or S2s to S5s (Table 1 and Fig 2), it is clear that a higher sequencing depth can cover missing core genes, even if their abundance is not as high as in S3s or S4s. Nevertheless, at sequencing above 33 million read pairs, sequencing depth seems to be no longer the limitation to increasing the accuracy of the assembly (S5 Fig) in terms of BUSCO results. For example, based on the relationship between the number of reads and BUSCO results for each library (S5 Fig), if we attempt to decrease "Missing" parts (BUSCO) to 15%, the optimal number of reads would be about 33 million read pairs (see the equation for trend-line of the "Missing" category in S5 Fig). This number is not a perfect estimate; however, it is a reasonable estimate from the observed trend.

### A comprehensive union

This study included libraries from three types of tissues: male-strobilus, leaf, and inner bark. These were collected from close but distinct genetic backgrounds, an asymmetrical number of

runs, and variable sample sizes. Considering that alternative splicing occurs in different conditions, e.g., tissues or treatment, we independently assembled each library, and then merged them into an integrated library. Using EvidentialGene as well as manual integration, the 49,758 sugi CJ3006NRE has resulted in a high confidence in this dataset. Most of these sequences were assembled independently in multiple libraries in the semi-manual process in the current study, as well as being selected by the EvidentialGene pipeline.

Repetitive sequences and isoforms co-exist in CJ3006NRE. Their compositions reflect the variation in the samples collected. Increases in accuracy and efforts toward completeness of the transcriptome data are important for sugi in the post-genomic era. On the other hand, the redundancy has been dealt with using two separate methods, the EvidentialGene pipeline and manual integration. There are two advantages to integration after assembly: all isoforms can be kept, and they can be used to validate contigs among libraries. If the assembly tools have a lower missassembly rate, it is possible to retain the isoforms for subsequent structural annotation, while the pseudomolecules remain available. Keeping the contigs assembled in at least two different libraries is preferable to overcome the missassembly due to randomized k-mer alignment processes. Because we assembled transcriptomes for each library with different k-mers (S3 Table) and only kept contigs which existed in multiple (existed at least two) libraries, the chance of producing the same missassembled contig would be low, but isoforms or gene families may exist in a part of the missassembly. Merging multiple assemblies is a step that should be standardized in *de novo* transcriptome assembly. By empirical results, both assemblers, Oases (by Velvet) and Trinity, produced a set of partially randomized contigs. We found this randomization to be a non-intended advantage; we used it to increase the reliability of our data by decreasing the chances of misassembly. Contig would be kept only if they could be found (or assembled) in at least two different assemblies from the same set of raw data, those assemblies being by different assemblers or by the same assembler with different parameters, and the raw data coming from the same dataset.

## Discarding contamination

Filtering out contamination is an important step when assembled contigs are to be used in downstream analysis. CJ3006NRE was selected based on the alignment results against the NCBI-NR database, and the taxa information of the database sequence was used in filtering criteria. In the current study, if any contigs were aligned with sequences of non- "eukaryote" origin, they were filtered out. As algae and fungi contaminations could theoretically be collected while sampling from sugi tissue, the term "eukaryote" may be too loose of a filter. "Euphyllophyta" and "Spermatophyta" are other potential thresholds, as they are more specific. However, considering that the accumulation of genomic data for conifers has not been established as well as for angiosperms, orthologs in sugi may have less homology that could be identified by BLAST. We used the "non-eukaryote" term as our negative threshold in order to discard the "non-Eukaryota" sequences, including those in Archaea, or Bacteria. Surprisingly, the reduction in the mapping rate of input reads was only less than 4%. Therefore, we suggest the use of CJ3006NRE, with 49,795 sequences, as a reference transcriptome, in place of CJ3006All, with 116,466 sequences, some of which showed similarity to "non-Eukaryota" sequences and might therefore be from contaminants.

## Increasing the accuracy of assembly

Based on the benchmark results, completeness, especially "single-copy," increased to up to 78.47% of CJ3006NRE from less than 45% of the S4s library (Fig 2). This suggests the integration process dramatically decreased the duplication and fragments. Following the ideas of

BUSCO [46], a higher number of duplicates may indicate a more erroneous assembly of a haplotype. However, the evaluation by BUSCO showed that our assembled contigs had less duplicates, suggesting that our data quality control effectively increased the quality of the assembly.

The more full-length cDNA is available, the higher the accuracy for subsequent annotation of the genomic sequences. We used both third-generation sequencing and full-length cDNA library downloaded from NCBI to validate accuracy of our data (Table 2). If we take ISO-Seq0215 (56,399 sequences) as the validation standard, and above 90% coverage of genes as the threshold, CJ3006NRE covered 35,741 (63.3%) of ISOSeq0215 sequences using only 12,231 (24.6%) of 49,785 sequences of the CJ3006NRE library (S9 Table). This demonstrates two points; at least 25% of CJ3006NRE could be validated by ISOSeq0215, and there were more CJ3006NRE sequences with potential full-length cDNA than in other cDNA resources. Overall, these results indicate that the methodology used in this study will be useful for assembling transcriptome data of non-model plant organisms.

## Contig numbers of all sugi genes

The total number of loci in sugi haplotypes is uncertain without genomic sequencing. As we collected the total RNA from male strobili, leaf, and bark, studies on other tissue types and conditions will be fruitful to complete this annotation. The discovery of more functional genes from different transcriptome datasets from different tissues or conditions is expected in the future. BUSCO estimated about 10.7% of the core gene has not yet been identified yet (Fig 2). Part of this 10.7% may include differences between the core gene of angiosperms and gymnosperms, which may become clearer with the development of more advanced sequencing technology, and genomic data of more gymnosperm species becoming available.

## Orthologous genes for pollen development

There were 74 orthologs selected from two model plants as candidates for male sterility genes (S10 Table). This list of orthologs was generated from 78 references (S11 Table). The similarity threshold to select these orthologs in CJ3006NRE proteome was as follows: E-value < 1e-20, and identity > = 40% based on the NCBI-BLASTP result, leading to the list of IDs of top two highest score for each candidate (S10 Table). The comparison of gene expression between male flower (MF) and inner bark and leaf (IBL) libraries suggested male sterility-related transcription factors (MYB80 and MYB35) differentially expressed in MF tissue (S12 Table). Focusing on downstream genes of MYB80 [59], we identified ortholog of a pectin methylesterase (VGD1), a glyoxal oxidase (GLOX1), and an A1 aspartic protease (UNDEAD). All of these genes are direct target of MYB80, and dysfunction of VGD1 and UNDEAD causes partial male sterility [59, 60]. Since the exact orthologs are unclear, the top two subjects were selected. Within each potential orthologs group, only one gene showed significantly higher expression in MF than in IBL (S12 Table). Although the downstream orthologs lack confirmation, the patterns of gene regulation in conifers may be different from the proposed model [59] of pollen development in *Arabidopsis*. It should also be noted that Phan *et al*. (2011) [59] compared mature and young floral bud tissues, while in the current study we compared male flower (MF) and inner bark and leaf (IBL) tissue for expression analysis.

## Expression differences between male-fertile and -sterile samples and *MS1* candidate gene

The time-series expression analysis of the paired male-fertile line (S3) and male-sterile line (S4) is expected to identify the temporal branch point of male sterility by observing the gene expression. In the libraries S3s and S4s, the suffixes "_rep1" to "_rep5" represent the five

sampling stages, from stage 1 to stage 5, respectively. In the rest of the accessions, the suffixes did not correlate with sampling stages. In the sample heatmap (S6 Fig), the major factor to cluster samples was highly related to sampling stage. Focusing on the time-series pair in libraries S3s and S4s, the paired accessions were bound in the same sub-clusters at the first, fourth, and fifth time points, but not at the second and third time points. In the clustering for S3s and S4s, S3s_rep2 was grouped with S3s_rep3, and S4s_rep2 and S4s_rep3 were not grouped with S3s_rep2 and S3s_rep3, respectively. This implied that the second sampling stage (11th Oct.) was the temporal branch point. However, we could not determine if the key gene had differential expression at the second sampling stage or if we observed a consequence of this difference, which may mean that the difference began at the time of the first sampling. Thus, the second and third sampling stages represent the key stages to examine differential analysis of male-fertile (S3s) and male-sterile (S4s) candidate genes. Tsubomura *et al.* (2016) also showed that gene expression differed between male-fertile (wild type) and male-sterile (*MS1* mutant) samples during developmental stage 4 to 6, which correspond to "Microspore mother cells enter meiosis" to "Callose wall surrounding tetrads degenerates and individual microspores released," respectively.

Using our datasets, we identified a candidate gene (CJt020762) for *MS1* [61], which has expected function in lipid transportation and frame-shift mutations in mutant alleles (*ms1*). The levels of expression over a time course were estimated for male-fertile S3s and -sterile S4s accessions (S7 Fig). At the first sampling stage (6th Oct.), both libraries showed low levels of expression. Then, from the second to fifth sampling time points, the expression in S3s was higher than in S4s, although the expression level decreased at the fourth and fifth sampling stage for both libraries. Microscopic images of the pollen at the fourth sampling stage (19th Oct.) showed microspores for S3s pollen, while in the male-sterile S4s, abnormalities were observed and no microspores were evident at this sampling stage (S7 Fig). This also suggested that the candidate gene functioned prior to the fourth sampling time point.

mRNA with a premature stop codon may be degraded by RNA surveillance mechanisms [62], which may result in the lower expression of CJt020762 for S4s, which display a premature stop codon due to a 4-bp deletion. According to Tsubomura *et al.* (2016), who categorized the gene expression of *C. japonica* male strobili into eight clusters along the developmental stage, the expression pattern of CJt020762 may be classified into C7 (genes expressed in the early and middle developmental stages). Unfortunately, the past gene expression studies by microarray [7, 63] were unable to find this candidate gene, as these microarrays did not have probes designed for the gene.

## Conclusion

In this work, we performed *de novo* assembly of *Cryptomeria japonica*, or sugi, by integrating transcriptome data from unequal runs of 10 cDNA libraries. These were constructed from multiple tissue types with slightly different genetic backgrounds within a short period. By using the public EvidentialGene pipeline and semi-manual integration, 49,795 reference transcripts were produced with high coverage (more than 90% of total reads mapped). Completeness of the transcriptome was saturated, with 33 million read pairs (6.6 Gb) per library. Based on the reference transcripts, we identified orthologous genes with a particular focus on pollen development. Expression differences between libraries were mostly clustered by tissue type. However, subtle difference between male-fertile and sterile samples could also be detected, and a candidate gene for *MS1* was identified. The reference transcript sequences and the potential SNPs identified in this study may be useful for future breeding and genetic research of this plant species.

## Supporting information

**S1 Table. RNA-Seq libraries sequenced by HiSeq from previous studies and the read mapping rate onto CJ3006NRE.** Accession number, Sample, Working ID, and Percentage (%) of mapped read onto CJ3006NRE.
(XLSX)

**S2 Table. DRA accession numbers of RNA-Seq libraries and their characteristics used in the current study.** WorkingID, Label, DDBJ Biosample accession number, Experiment, Number of reads or read pairs, Sequencing Technology, Tissue, Sample name, and Sample title.
(XLSX)

**S3 Table. Statistics of assembled libraries.** Two assembly tools (Oases and Trinity) were used and the k-mer size, the number of contigs, N50 length, and sum of the total contig length are summarized for each assembly.
(XLSX)

**S4 Table. The annotation of CJ3006NRE.** Each sequence in CJ3006NRE was annotated in terms of predicted enzyme function (E2P2), transcription factor (TF), transposable element (TE), isoforms, and SNP markers. Sugi SNPs are based on Uchiyama et al. 2013. GSE64663 was published by Tsubomura *et al*. 2016, while GSE95616, and GSE95618 were published by Mishima *et al*. 2018. Orthologues were identified from *Arabidopsis thaliana*, *Oryza sativa*, *Picea abies* and *Pinus taeda* proteome sequences downloaded from PLAZA web site (https:// bioinformatics.psb.ugent.be/plaza/).
(XLSX)

**S5 Table.** Major components of transposable elements for (a) CJ3006NRE and other libraries, and (b) a RepeatMasker output example for CJ3006NRE. The major components (SINEs, LINEs, LTR—Ty1/Copia, LTR—Gypsy/DIRS1, and DNA transposons) were extracted from the RepeatMasker output for each library.
(XLSX)

**S6 Table. The mapping rate of raw reads against different reference sequences.** Self: the reference consisted of the contigs assembled by the same library of raw reads; CJ3006All: the contigs set of CJ3006All without filtering; CJ3006NRE: the contigs set of CJ3006NRE (the CJ3006All with only matches to eukaryotes in NCBI-NR); Only EviGene: CJ3006NRE without the manual integrated dataset; FLcDNA: the full-length cDNA set download from NCBI on September 13, 2018; ISOSeq0215hq: High-quality subset of ISO-Seq data
(XLSX)

**S7 Table. Differentially expressed transcripts between "male flower" (MF) and "inner bark and leaf" (IBL) tissue sets with a focus on transcription factors.** Differential expression was analyzed by the slueth tool showing target_id, pval, qval, and b column. target_id: CJ3006NRE contig ID, pval: the Wald test FDR-adjusted p-value using Benjamini-Hochberg, qval: the p-value adjusted for multiple test correction, b: the natural log of the fold change between conditions. TF_acc and TF_name show transcription factor accession numbers in the Pfam database and their names, respectively.
(XLSX)

**S8 Table. Number of specific variations detected for each parental line.** Accession-specific variants were classified using the "isec" command in bcftools.
(XLSX)

**S9 Table. Mutual number of sequences (CJ3006NRE/cDNA source) under different coverage thresholds.** CJ3006NRE contigs were BLASTed against each cDNA source and coverage was calculated for each coverage threshold.
(XLSX)

**S10 Table. The orthologous candidates involved in microsporogenesis.**
(XLSX)

**S11 Table. References used for selecting orthologous candidates involved in microsporogenesis.**
(XLSX)

**S12 Table. Differential expression of MYB80-related genes of male flower (MF) against inner bark and leaf (IBL) tissues.** Differential expression analyzed by the slueth tool. pval: the Wald test FDR-adjusted p-value using Benjamini-Hochberg, qval: the p-value adjusted for multiple test correction, b: the natural log of the fold change of male strobili to leaf and inner bark gene expression.
(XLSX)

**S1 Fig. The pedigree of *C. japonica* accessions.** Pedigrees for the analysis of a) *MS1*, b) *MS2*, c) *MS3*, and d) *MS4*. Genotypes for male sterility genes are indicated by color patterns, where solid colors and gradient colors correspond to male-fertile and male-sterile genotypes, respectively. Inside shapes are the names of RNA-Seq library working IDs (Table 1).
(PNG)

**S2 Fig. Workflow of assembly.** The general workflow of assembly and the workflow of the semi-manual assembly (module-1, module-2, and modle-3).
(PNG)

**S3 Fig. Diagram for identifying group-specific variants.** This bitmap pattern was used in the "bcftools isec" command to select group specific variants.
(PNG)

**S4 Fig. Relationship between pairwise relatedness (x-axis) and Pearson's product-moment correlation coefficient (y-axis) of gene expression.** Each dot indicates a pairwise comparison of RNA-Seq libraries. MF: Male flower, IBL: inner bark and leaf, MFIBL: male flower, inner bark, and leaf.
(PPTX)

**S5 Fig. The number of reads and the BUSCO benchmark results.** The x-axis indicates the number of read pairs in each library, while the y-axis indicates BUSCO results ratios for Complete, Duplicated, Fragments, Missing, and Singles.
(PDF)

**S6 Fig. Heatmap for the correlation of gene expression between S3s (male-fertile) and S4s (male-sterile) libraries.** Correlations were analyzed via a time-series of sampling, dated from 6th Oct to 24th Oct.
(PPTX)

**S7 Fig. Gene expression (tags per million) of CJt020762, a candidate gene of *MS1*.** Microscopic images (400x) of pollen, taken on 19th Oct. 2011, for S3s and S4s are embedded in the graph.
(PPTX)

**S1 Appendix. Examples of command lines for variant calling.**
(TXT)

## Acknowledgments

The authors would like to thank Shinji Itoo for producing a part of the materials used in the current study. The Arboretum and Nursery Office (FFPRI) was involved in the maintenance of the *C. japonica* nursery. Computations were mostly performed on a supercomputer at AFFRIT, MAFF, Japan. The manuscript was proofread by Enago for grammar and spelling.

## Author Contributions

**Conceptualization:** Fu-Jin Wei.

**Data curation:** Saneyoshi Ueno, Tokuko Ujino-Ihara.

**Formal analysis:** Fu-Jin Wei, Saneyoshi Ueno.

**Funding acquisition:** Saneyoshi Ueno, Yoshinari Moriguchi.

**Investigation:** Fu-Jin Wei, Saneyoshi Ueno.

**Methodology:** Fu-Jin Wei, Tokuko Ujino-Ihara.

**Project administration:** Saneyoshi Ueno.

**Resources:** Saneyoshi Ueno, Maki Saito, Yuumi Higuchi, Satoko Hirayama, Junji Iwai, Tetsuji Hakamata.

**Supervision:** Saneyoshi Ueno.

**Validation:** Saneyoshi Ueno.

**Visualization:** Saneyoshi Ueno.

**Writing – original draft:** Fu-Jin Wei, Saneyoshi Ueno.

**Writing – review & editing:** Fu-Jin Wei, Saneyoshi Ueno, Tokuko Ujino-Ihara, Yoshihiko Tsumura.

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
