## [Decision Letter · Decision Letter 0]

14 May 2020

PONE-D-20-09138

Inspecting abundantly expressed genes in male strobili in sugi (Cryptomeria japonica D. Don) via a highly accurate cDNA assembly

PLOS ONE

Dear Dr. Ueno,

Thank you for submitting your manuscript to PLOS ONE. After careful consideration, we feel that it has merit but does not fully meet PLOS ONE’s publication criteria as it currently stands. Therefore, we invite you to submit a revised version of the manuscript that addresses the points raised during the review process.

The reviewers have pointed to merit in this report, yet they have raised serious concerns on the suitability of this report to receive scientific peer review in its current form.  Please address every concern of the reviewers and consider a complete rewrite of the manuscript including its scope. It is also recommended by the editor that a native English speaker thoroughly edit the report prior to resubmission..

We would appreciate receiving your revised manuscript by Jun 28 2020 11:59PM. To enhance the reproducibility of your results, we recommend that if applicable you deposit your laboratory protocols in protocols.io, where a protocol can be assigned its own identifier (DOI) such that it can be cited independently in the future. For instructions see: http://journals.plos.org/plosone/s/submission-guidelines#loc-laboratory-protocols

We look forward to receiving your revised manuscript.

Kind regards,

F. Alex Feltus, Ph.D.

Academic Editor

PLOS ONE

Journal Requirements:

Reviewers' comments:

Reviewer's Responses to Questions

**Comments to the Author**

1. Is the manuscript technically sound, and do the data support the conclusions?

Reviewer #1: Partly

Reviewer #2: Partly

2. Has the statistical analysis been performed appropriately and rigorously? 

Reviewer #1: Yes

Reviewer #2: N/A

3. Have the authors made all data underlying the findings in their manuscript fully available?

Reviewer #1: Yes

Reviewer #2: No

4. Is the manuscript presented in an intelligible fashion and written in standard English?

Reviewer #1: Yes

Reviewer #2: No

5. Review Comments to the Author

Reviewer #1: The manuscript entitled "Inspecting abundantly expressed genes in male strobili in sugi (Cryptomeria japonica D. Don) via a highly accurate cDNA assembly " by Wei et al. present the results of de novo assembly of Cryptomeria japonica in male strobili, by integrating transcriptome data.

Unfortunately, the manuscript as it is suffers of some serious flaws as well as other issues that prevent it to be recommended for publication in its present form.

I suggest revisions on the following issues

1.In this study, authors was intended to construct a high-quality cDNA assembly on Cryptomeria ,and show some specificity of genes in male strobili for male sterility.

However authors should pay more attention to comparison between those from fertile and sterile clones under purpose, or other male sterile genes (ms1,ms2,ms3,ms4) from other Cryptomeria male sterile clones, rather than those from different tissue (inner bark & leaf). 

Moreover it was not clear when tissue to construct libraries from Inner bark & leaf was collected in this manuscript. Of course, gene expression  of male or female cone is differed by developmental stage. Similar, gene expression profile from inner bark and leaf is affected by every season.

Cover rate derived from BUSCO is affected by difference between seasons, or also tissues, including that not collected in this study (although authors also showed it in this manuscript). Discussion including differences  between angiosperms and gymnosperms in cover rate is a slightly unsuitable in this manuscript.　

In the first place, gene expression derived from different season and tissues in Cryptomeria have been already published(Mishima et al. 2014,Nose & Watanabe 2014, Mishima et al. 2018 and so on).Then, discussion of the results with these previous studies would futher enhance the manuscript.

2.Comparison of transposable elements, transcription factors,CSD etc., was performed with those of Arabidopsis an Oryza in this study. However more species will be required. Because a conifer species, Cryptomeria is evolutionary distant  from both angiosperms species, and database from angiosperm woody plants and gymnosperm species have been already prepared.

3.Authors almost never discuss for variant calling.

Relatedness between gene expression and variant with comparison of male sterile genes (ms1,ms2,ms3,ms4) is  needed to investigate more.

Point out minor comments

1.Difference between CJ3006ALLとCJ3006NER should show in “materials & method, not legend in Fig.2.

2.It is not clear because name of database (query) was not noted.

example；

-In P9:Explanation of “ISOSeq215” need to note.

-In P10L190, Is "FLcDNA" same with ”CJFLcDNA" in P8L154 ?

-In P26L547, what is name ?, ”CJFLcDNA"？

3.Unify abbreviation of male strobili used.MS and MF is mixed in the manuscript.

4.Could you increase the Figure resolution.I can not read character in figure (only my PC?)

Reviewer #2: The manuscript presents the transcriptome of Cryptomeria japonica, a conifer species that has been extensively used in forestry plantations in Japan. In terms of research, the authors have done a considerable amount of work and a number of different analyses on the data. The transcriptome will provide an excellent resource for future studies with this species. However, I felt the manuscript itself needed a lot more work before it would be ready for publication. Not only the manuscript contains several awkward passages and phrases which make it difficult to follow, but also there are many places where the text doesn't provide the detail and clarity required for understanding. I have attached a file in which I highlighted many parts of the text that I found not clear. Please be aware that I did not include grammar errors (which should also be corrected), as I tried not to focus on language problems but the scientific contents. The attached file contains also many concerns on varying aspects of the work, the most important ones being:

(1) The title suggests that this paper is evaluating differential expression, but the abstract, introduction, and conclusions do not reflect that. Instead, the paper is focused on generating a transcriptome assembly and evaluating its quality. Therefore, I strongly recommend a change of the title.

(2) Conclusions are not particularly convincing, even if they are not wrong. The authors repeatedly praised having high confidence in their assembly and the fact that studies in functional genomics and evolutionary biology will benefit from this transcriptome. That is too generic in my opinion, and I recommend concluding something that fits better with the work done.

(3) The results are not clearly presented. In fact, they felt extremely confusing to me. Even though I tried my best to understand them, figures and text need a more exhaustive explanation to be comprehensible. Some important terms needed to understand the results were not explained in detail when they were first mentioned. Figure legends need additional work to make them stand alone.

(4) The discussion makes comparisons that felt inappropriate to me. The results obtained for the transcriptome of Sugi (gymnosperm) are compared with different angiosperm species. I think these are too distantly related and cannot be compared easily. Actually, in the introduction it is mentioned that there are few other gymnosperms with genome available: why the results were not compared to these more closely-related plants instead?

(5) This is more of a methods paper in my opinion, but the different steps of pipeline (which is shown in Fig. S4) were not easy to follow. I think the pipeline should be more clearly explained in the main text.

(6) The research material should be available upon submission to ensure reproducibility. Data should be submitted to GenBank in both raw and assembled forms. Homemade scripts are used but not available either.

6. PLOS authors have the option to publish the peer review history of their article (what does this mean?). If published, this will include your full peer review and any attached files.

Reviewer #1: No

Reviewer #2: No

---

## [Author Response · Author response to Decision Letter 0]

13 Nov 2020

RE: PONE-D-20-09138

Inspecting abundantly expressed genes in male strobili in sugi (Cryptomeria japonica D. Don) via a highly accurate cDNA assembly

Dear Dr. Feltus,

I am pleased to submit a revised version of our article entitled “Construction of a reference transcriptome for the analysis of male sterility in sugi (Cryptomeria japonica D. Don) focusing on MALE STERILITY 1 (MS1)”. Following the helpful comments of the reviewers, we have re-written our manuscript and hereby re-submit it with our responses to the reviewers.

We hope that our revised manuscript is now acceptable for publication in PLOS ONE, and look forward to hearing from you at your earliest convenience.

Yours sincerely,

Saneyoshi Ueno, Ph.D.

Department of Forest Molecular Genetics and Biotechnology, Forestry and Forest Products Research Institute, Forest Research and Management Organization, 1 Matsunosato, Tsukuba, Ibaraki 305-8687, Japan.

E-mail: saueno@ffpri.affrc.go.jp, Phone: +81-29-829-8261

Reply to reviewer#1

We thank the reviewer for their comments, which have greatly improved the manuscript. Our replies to these comments follow here:

1.In this study, authors was intended to construct a high-quality cDNA assembly on Cryptomeria ,and show some specificity of genes in male strobili for male sterility.

However authors should pay more attention to comparison between those from fertile and sterile clones under purpose, or other male sterile genes (ms1,ms2,ms3,ms4) from other Cryptomeria male sterile clones, rather than those from different tissue (inner bark & leaf). 

Reply 1:

In the revised version, we have added analysis focusing on the expression differences between sterile and fertile libraries of male strobili (Page 25, line 530 – Page 26, line 550 in the text version without track changes). Unfortunately, comparison between other male sterility genes (ms1, ms2, ms3, and ms4) were impossible, due to the large effects of tissue types. In the current study, sampled tissue types differed for male sterility genes (ms1, ms2, ms3, and ms4).

Moreover it was not clear when tissue to construct libraries from Inner bark & leaf was collected in this manuscript. Of course, gene expression of male or female cone is differed by developmental stage. Similar, gene expression profile from inner bark and leaf is affected by every season.

Reply 2:

We have clarified the sampling date for each sample in S2 Table. 

Cover rate derived from BUSCO is affected by difference between seasons, or also tissues, including that not collected in this study (although authors also showed it in this manuscript). Discussion including differences between angiosperms and gymnosperms in cover rate is a slightly unsuitable in this manuscript.　

Reply 3:

Thank you for your comment. We agree that the expressed genes vary among tissue types and seasons; the evaluation by BUSCO is carried out here to verify the completeness of the dataset CJ3006NRE, of which the number of reads sampled from a library is the most important factor (S5 Fig). We have deleted the discussion of the comparison between angiosperms and gymnosperms.

In the first place, gene expression derived from different season and tissues in Cryptomeria have been already published(Mishima et al. 2014,Nose & Watanabe 2014, Mishima et al. 2018 and so on).Then, discussion of the results with these previous studies would futher enhance the manuscript.

Reply 4:

We have downloaded RNA-Seq reads from Mishima et al. (2018) from NCBI-SRA and mapped them onto our dataset CJ3006NRE (S1 Table). Microarray probe sequences by Tsubomura et al. (2016) were also compared against CJ3006NRE (S4 Table). Discussion was added regarding the expression differences among fertile and sterile male accessions (page 26, lines 546-550) referencing Tsubomura et al. (2016). Unfortunately, the experimental design for the current study was not suited to discussion of the seasonal expression difference.

2.Comparison of transposable elements, transcription factors,CSD etc., was performed with those of Arabidopsis an Oryza in this study. However more species will be required. Because a conifer species, Cryptomeria is evolutionary distant from both angiosperms species, and database from angiosperm woody plants and gymnosperm species have been already prepared.

Reply 5:

Although we also agree with this analysis, at the request of reviewer #2, we have deleted the comparison between gymnosperms and angiosperms. Instead, we compared the sugi transcriptome with closely related coniferous species, and identified orthologous genes by the reciprocal best blast hit method (page 10, lines 196-201, and page 14, lines 304-307).

3.Authors almost never discuss for variant calling.

Relatedness between gene expression and variant with comparison of male sterile genes (ms1,ms2,ms3,ms4) is needed to investigate more.

Reply 6:

We have added analysis of variant calling and gene expression (page 11, lines 243-252, and page19, lines 408-418). We found that with higher relatedness, there was higher correlation of gene expression. However, this correlation likely only reflects the current experimental design.

Point out minor comments

1.Difference between CJ3006ALLとCJ3006NER should show in “materials & method, not legend in Fig.2.

Reply 7:

We have defined CJ3006ALL and CJ3006NRE on page 8, lines 174-177, and on page 13, lines 279-281.

2.It is not clear because name of database (query) was not noted.

example；

-In P9:Explanation of “ISOSeq215” need to note.

-In P10L190, Is "FLcDNA" same with ”CJFLcDNA" in P8L154 ?

-In P26L547, what is name ?, ”CJFLcDNA"？

Reply 8:

We have defined ISOSeq215 and CJ_FLcDNA on page 10, lines 220 and 222, respectively. We unified the terms FLcDNA and CJFLcDNA as CJ_FLcDNA.

3.Unify abbreviation of male strobili used.MS and MF is mixed in the manuscript.

Reply 9:

We have unified these abbreviations, and use only the term MF in the revised version of the manuscript.

4.Could you increase the Figure resolution.I can not read character in figure (only my PC?)

Reply 10:

We have increased the figure resolution in the revised manuscript.

 

 

Reply to reviewer#2

The authors thank the reviewer for their comments, which have greatly improved our manuscript. Please see our replies to these comments below:

Reviewer #2: The manuscript presents the transcriptome of Cryptomeria japonica, a conifer species that has been extensively used in forestry plantations in Japan. In terms of research, the authors have done a considerable amount of work and a number of different analyses on the data. The transcriptome will provide an excellent resource for future studies with this species. However, I felt the manuscript itself needed a lot more work before it would be ready for publication. Not only the manuscript contains several awkward passages and phrases which make it difficult to follow, but also there are many places where the text doesn't provide the detail and clarity required for understanding. I have attached a file in which I highlighted many parts of the text that I found not clear. Please be aware that I did not include grammar errors (which should also be corrected), as I tried not to focus on language problems but the scientific contents. The attached file contains also many concerns on varying aspects of the work, the most important ones being:

(1) The title suggests that this paper is evaluating differential expression, but the abstract, introduction, and conclusions do not reflect that. Instead, the paper is focused on generating a transcriptome assembly and evaluating its quality. Therefore, I strongly recommend a change of the title.

Reply 1:

We have rephrased almost all of the text that you mentioned here. We have also changed the title to “Construction of a reference transcriptome for the analysis of male sterility in sugi (Cryptomeria japonica D. Don) focusing on MALE STERILITY 1 (MS1)”. The revised manuscript was proofread by an English-language editor at Enago for grammar and spelling.

(2) Conclusions are not particularly convincing, even if they are not wrong. The authors repeatedly praised having high confidence in their assembly and the fact that studies in functional genomics and evolutionary biology will benefit from this transcriptome. That is too generic in my opinion, and I recommend concluding something that fits better with the work done.

Reply 2:

Per the reviewer’s comments, we have rewritten the conclusion section.

(3) The results are not clearly presented. In fact, they felt extremely confusing to me. Even though I tried my best to understand them, figures and text need a more exhaustive explanation to be comprehensible. Some important terms needed to understand the results were not explained in detail when they were first mentioned. Figure legends need additional work to make them stand alone.

Reply 3:

We have updated the manuscript, and clearly explain all terms at their first appearance in the text. We also have added explanations to figure legends.

(4) The discussion makes comparisons that felt inappropriate to me. The results obtained for the transcriptome of Sugi (gymnosperm) are compared with different angiosperm species. I think these are too distantly related and cannot be compared easily. Actually, in the introduction it is mentioned that there are few other gymnosperms with genome available: why the results were not compared to these more closely-related plants instead?

Reply 4:

Most of the comparisons with angiosperms have been deleted in the revised version. Instead, comparison to gymnosperm transcriptomes has been added (page 9, line 196-201, and page 14, line 304-307). Unfortunately, detailed gene annotation for gymnosperm transcriptomes is limited. We therefore were only able to identify orthologues of C. japonica genes in other gymnosperm transcriptomes.

(5) This is more of a methods paper in my opinion, but the different steps of pipeline (which is shown in Fig. S4) were not easy to follow. I think the pipeline should be more clearly explained in the main text.

Reply 5:

In the revised version, we have added further explanation of the methods, and a pipeline script has been uploaded to the Bitbucket repository.

(6) The research material should be available upon submission to ensure reproducibility. Data should be submitted to GenBank in both raw and assembled forms. Homemade scripts are used but not available either.

Reply 6:

Both the raw reads and assembled contigs are now registered, and accession numbers are provided in the text (page 27, lines: 585). Scripts used in the analysis have also been uploaded to Bitbucket (https://bitbucket.org/saueno1/masker/src/master/) (page 28, line 589).

---

## [Decision Letter · Decision Letter 1]

9 Dec 2020

PONE-D-20-09138R1

Construction of a reference transcriptome for the analysis of male sterility in sugi (Cryptomeria japonica D. Don) focusing on MALE STERILITY 1 (MS1)

PLOS ONE

Dear Dr. Ueno,

Thank you for submitting your manuscript to PLOS ONE. After careful consideration, we feel that it has merit but does not fully meet PLOS ONE’s publication criteria as it currently stands. Therefore, we invite you to submit a revised version of the manuscript that addresses the points raised during the review process.

While the deepest concerns of the reviewers have been addressed, please respond the remining comments that should greatly improve the report. 

We look forward to receiving your revised manuscript.

Kind regards,

F. Alex Feltus, Ph.D.

Academic Editor

PLOS ONE

Reviewers' comments:

Reviewer's Responses to Questions

**Comments to the Author**

1. If the authors have adequately addressed your comments raised in a previous round of review and you feel that this manuscript is now acceptable for publication, you may indicate that here to bypass the “Comments to the Author” section, enter your conflict of interest statement in the “Confidential to Editor” section, and submit your "Accept" recommendation.

Reviewer #1: All comments have been addressed

Reviewer #2: (No Response)

2. Is the manuscript technically sound, and do the data support the conclusions?

Reviewer #1: Partly

Reviewer #2: Yes

3. Has the statistical analysis been performed appropriately and rigorously? 

Reviewer #1: Yes

Reviewer #2: Yes

4. Have the authors made all data underlying the findings in their manuscript fully available?

Reviewer #1: Yes

Reviewer #2: Yes

5. Is the manuscript presented in an intelligible fashion and written in standard English?

Reviewer #1: Yes

Reviewer #2: Yes

6. Review Comments to the Author

Reviewer #1: The manuscript presents the construction of a reference transcriptome for the analysis of male sterility in in Cryptomeria japonica.

The revised version has been amended based on the comments, but I think it still needs some improvement before it would be ready for publication.

I agree that the comparison　between other male sterility genes (ms1, ms2, ms3, and ms4) 　were impossible　, due to the large effects of tissue types as stated by authors.Similarly, I also agree that the results of analysis of variant calling and gene expression would only reflect the current experimental design.

Althougth, in the revised version, focusing on the expression differences of ms1 were added, it would be possible to see the relationship between variant and gene expression within the ms1 group, and this might clarify the ms1 male sterility gene feature. In addition, data such as variants in orthologous genes for pollen development would help to compare four male sterility genes (ms1, ms2, ms3, and ms4).

However, due to changes in title, conclusions, and so on, unlike the original, it seems that too mach discussion centered on ms1 may deviate from the purpose of the paper (construction of the reference transcriptome), so it might be better to leave out the discussion of gene expression including differences in expression between tissues.

Reviewer #2: The authors have made a considerable load of work in this revised version of the manuscript. Overall, I am very satisfied with most of their corrections. I found the text clearly explained and much easier to follow compared to the previous version. However, after rereading the manuscript, I still have the following minor comments:

- Line 24: The sentence "Transcriptomics is one approach that can address the deficiency", to which "deficiency" it refers? Maybe it should say "this" instead of "the".

- Line 28: The code "CJ3006NRE" should be removed from here as its meaning is not clear at this point in the abstract. Its meaning is clearer on Line 36, so I would keep it only there.

- Line 44: Instead of "a primary reasons" it should say "a primary reason".

- Line 47: I am not sure what "23 male-sterile trees" means. Do you mean there are up to 23 male-sterile tree varieties that could be used for breeding?

- Lines 50-52: The way this sentence is written is a bit confusing to me, as it makes me think that "to more precisely pinpoint the genetic variation or genes related to male sterility" are the goals of your study.

- Line 57: This sentence should be written as "So far, only seven gymnosperm genomes have been published".

- Line 87: I think Supplementary Materials are more commonly written as "Fig. S1" or "Figure S1" instead of "S1 Fig." (and the same for other Supplementary Figures and Tables throughout the text).

- Line 87-92: The codes "T5_normalMIX_ms1", "T5_sterileMIX_ms1", "S3T67_normalMIX_ms1", "S3T67_sterileMIX_ms1", and "Ooi-7" do not match the ones shown in Fig. S1.

- Line 125: It should better say "Sampling bias" instead of "bias in the samples".

- Lines 168-177: These 3 paragraphs are each too short. I recommend having only one paragraph here.

- Lines 176-177: Given that this is the first time in the main text that the code "CJ3006NRE" is cited, I think this sentence should be clearer in saying that "CJ3006NRE" is the name of your integrated library which was used for downstream analyses.

- Lines 179-201: Text is oversplit. I suggest having one or two paragraphs here.

- Line 205: What does "as done for assembly" mean?

- Line 267: The legend in Fig. 1(a) should also say "male strobili" and "inner bark or leaf" under the tissue label.

- Lines 263-265: In my opinion, the last sentence "The purpose of ..." is not 100% needed.

- Lines 292-293: Text is oversplit. I would keep these two lines at the end of the previous paragraph.

- Line 327: Add the names of the two other cDNA sources ("ISOSeq0215", "ISOSeq0215hq").

- Line 571: I feel that the sentence "focusing on male sterility in gymnosperms" should not be written here (maybe it should be written later in the paragraph or just be deleted).

- Line 586: The first sentence does not say where the accessions can be found. My guess (after reading the second sentence) is that these accessions are somewhere within the ForestGEN database. However, I could not find them anywhere after searching for a while. I still think that the data should be better uploaded to NCBI as it is a more standard database. RNA-seq data in FASTQ format can be deposited at the NCBI Sequence Read Archive (SRA), whereas transcriptome assemblies in FASTA can be deposited at DDBJ/EMBL/GenBank on the TSA (Transcriptome Shotgun Assemblies) section. If for some reason the authors are required to upload their data into ForestGEN, I would still suggest uploading to NCBI as well.

7. PLOS authors have the option to publish the peer review history of their article (what does this mean?). If published, this will include your full peer review and any attached files.

Reviewer #1: No

Reviewer #2: No

---

## [Author Response · Author response to Decision Letter 1]

9 Jan 2021

Reply to reviewer#1

Thank you for your comments, which are suggestive for our future research. Our reply to your comments follows:

Reviewer #1: The manuscript presents the construction of a reference transcriptome for the analysis of male sterility in in Cryptomeria japonica.

The revised version has been amended based on the comments, but I think it still needs some improvement before it would be ready for publication.

I agree that the comparison　between other male sterility genes (ms1, ms2, ms3, and ms4) 　were impossible　, due to the large effects of tissue types as stated by authors. Similarly, I also agree that the results of analysis of variant calling and gene expression would only reflect the current experimental design.

Althougth, in the revised version, focusing on the expression differences of ms1 were added, it would be possible to see the relationship between variant and gene expression within the ms1 group, and this might clarify the ms1 male sterility gene feature. In addition, data such as variants in orthologous genes for pollen development would help to compare four male sterility genes (ms1, ms2, ms3, and ms4).

Reply:

Thank you for your comment. We are sorry that experimental design in this study did not focus on gene expression. However in our companion paper (Hasegawa et al. 2021), we analyzed expressional differences ‘within ms1 group,’ comparing S3s (MS1/ms1) and S4s (ms1/ms1) library. We partly agree that variants in orthologous genes for pollen development would help compare four male sterile genes. The difficulty in such comparison in sugi is that the species is highly heterogeneous, which makes complicated such kind of comparison. In our opinion, comparison is effective between relatively homozygous lines such as those within mapping family.

However, due to changes in title, conclusions, and so on, unlike the original, it seems that too mach discussion centered on ms1 may deviate from the purpose of the paper (construction of the reference transcriptome), so it might be better to leave out the discussion of gene expression including differences in expression between tissues.

Reply:

Because we have changed the title based on a comment by reviewer #2, and because sub-topic of the current study was to focus on the MS1 related genes, we would like to keep the discussion section including the expressional analysis.

 

Reply to reviewer#2

Reviewer #2: The authors have made a considerable load of work in this revised version of the manuscript. Overall, I am very satisfied with most of their corrections. I found the text clearly explained and much easier to follow compared to the previous version. However, after rereading the manuscript, I still have the following minor comments:

Reply:

Thank you for your comments and patient review, which greatly improves our manuscript. Our reply to your minor comments follows.

- Line 24: The sentence "Transcriptomics is one approach that can address the deficiency", to which "deficiency" it refers? Maybe it should say "this" instead of "the".

Reply:

Thank you for your comment. We have changed the text accordingly.

- Line 28: The code "CJ3006NRE" should be removed from here as its meaning is not clear at this point in the abstract. Its meaning is clearer on Line 36, so I would keep it only there.

Reply:

Thank you for your comment. We have removed the code accordingly.

- Line 44: Instead of "a primary reasons" it should say "a primary reason".

Reply:

Thank you for your comment. We have changed the text accordingly.

- Line 47: I am not sure what "23 male-sterile trees" means. Do you mean there are up to 23 male-sterile tree varieties that could be used for breeding?

Reply:

Thank you for your comments. At least 23 trees had been verified as genetic male-sterile trees by artificial crossing by 2010. These trees are recessive homozygote and do not produce pollen and can be used for breeding. The manuscript text has been modified slightly as follows: “as at least 23 male-sterile trees [4], each with one of the four independent recessive alleles (ms1, ms2, ms3, and ms4) responsible for male sterility, has been discovered and could be used for breeding.” 

- Lines 50-52: The way this sentence is written is a bit confusing to me, as it makes me think that "to more precisely pinpoint the genetic variation or genes related to male sterility" are the goals of your study.

Reply:

Thank you for your comment. We have modified the text as follows: “Recent advances in technology have revealed more details of these loci by a functional genomics study via transcriptomics approach to more precisely pinpoint the genetic variation or genes related to male sterility [6,7], as a reference genome sequence of sugi is not currently available.”

- Line 57: This sentence should be written as "So far, only seven gymnosperm genomes have been published".

Reply:

Thank you for your comment. We have modified the text accordingly.

- Line 87: I think Supplementary Materials are more commonly written as "Fig. S1" or "Figure S1" instead of "S1 Fig." (and the same for other Supplementary Figures and Tables throughout the text).

Reply:

Thank you for your comment. We followed the instruction of the PlosOne website (https://journals.plos.org/plosone/s/supporting-information ) for the designation of supplemental information.

- Line 87-92: The codes "T5_normalMIX_ms1", "T5_sterileMIX_ms1", "S3T67_normalMIX_ms1", "S3T67_sterileMIX_ms1", and "Ooi-7" do not match the ones shown in Fig. S1.

Reply:

Thank you for your comment. The S1 Fig described the pedigrees used in the current study. The codes "T5_normalMIX_ms1", "T5_sterileMIX_ms1", "S3T67_normalMIX_ms1", and "S3T67_sterileMIX_ms1" are names of RNA-Seq libraries, abbreviated as S3s, S4s, S5s, and S6s, respectively. The abbreviations (working IDs) of each library were included in S1 Fig. ‘Ooi-7’ is a heterozygous (Ms1/ms1) tree, from which RNA was extracted. No pedigree information was included in S1 Fig. for ‘Ooi-7’.

- Line 125: It should better say "Sampling bias" instead of "bias in the samples".

Reply:

Thank you for your comment. We have modified the text accordingly.

- Lines 168-177: These 3 paragraphs are each too short. I recommend having only one paragraph here.

Reply:

Thank you for your comment. These three paragraphs are now combined accordingly.

- Lines 176-177: Given that this is the first time in the main text that the code "CJ3006NRE" is cited, I think this sentence should be clearer in saying that "CJ3006NRE" is the name of your integrated library which was used for downstream analyses.

Reply:

Thank you for your comment. We have modified the text accordingly as “Sequences matching eukaryotes were remained as “CJ3006NRE,” which is the name of integrated library used for downstream analyses.”

- Lines 179-201: Text is oversplit. I suggest having one or two paragraphs here.

Reply:

Thank you for your comment. We have combined paragraphs into one.

- Line 205: What does "as done for assembly" mean?

Reply:

Thank you for your comment. We have deleted the phrase, because we think that it has no meaning.

- Line 267: The legend in Fig. 1(a) should also say "male strobili" and "inner bark or leaf" under the tissue label.

Reply:

Thank you for your comment. We have modified the figure legend accordingly.

- Lines 263-265: In my opinion, the last sentence "The purpose of ..." is not 100% needed.

Reply:

We agree to your comment. We have deleted the last sentence.

- Lines 292-293: Text is oversplit. I would keep these two lines at the end of the previous paragraph.

Reply:

Thank you for your comment. We have combined the paragraph into one accordingly.

- Line 327: Add the names of the two other cDNA sources ("ISOSeq0215", "ISOSeq0215hq").

Reply:

Thank you for your comment. We have added the names of other cDNA sources as “we compared the coverage between CJ3006NRE and three other cDNA sources, full-length cDNA by Sanger sequencing (CJ_FLcDNA) and full-length cDNA by ISO-Seq in our own laboratory (ISOSeq0215 and ISOSeq0215hq).”

- Line 571: I feel that the sentence "focusing on male sterility in gymnosperms" should not be written here (maybe it should be written later in the paragraph or just be deleted).

Reply:

Thank you for your comment. We have deleted the phrase accordingly.

- Line 586: The first sentence does not say where the accessions can be found. My guess (after reading the second sentence) is that these accessions are somewhere within the ForestGEN database. However, I could not find them anywhere after searching for a while. I still think that the data should be better uploaded to NCBI as it is a more standard database. RNA-seq data in FASTQ format can be deposited at the NCBI Sequence Read Archive (SRA), whereas transcriptome assemblies in FASTA can be deposited at DDBJ/EMBL/GenBank on the TSA (Transcriptome Shotgun Assemblies) section. If for some reason the authors are required to upload their data into ForestGEN, I would still suggest uploading to NCBI as well.

Reply:

Thank you for your comment. We have added the name of the depository as “Sequences and assembled transcripts presented in the current study have been deposited at DDBJ/EMBL/GenBank with accession numbers DRR174638-DRR174656 and ICQT01000001-ICQT01048643, respectively.” We had registered these entries when we submitted the previous version of our manuscript, and have made these accessions public as can be verified from examples bellow: 

http://getentry.ddbj.nig.ac.jp/getentry/na/ICQT01000001/?filetype=html

 

http://trace.ddbj.nig.ac.jp/DRASearch/submission?acc=DRA006304

http://trace.ddbj.nig.ac.jp/DRASearch/submission?acc=DRA008245

---

## [Decision Letter · Decision Letter 2]

3 Feb 2021

Construction of a reference transcriptome for the analysis of male sterility in sugi (Cryptomeria japonica D. Don) focusing on MALE STERILITY 1 (MS1)

PONE-D-20-09138R2

Dear Dr. Ueno,

We’re pleased to inform you that your manuscript has been judged scientifically suitable for publication and will be formally accepted for publication once it meets all outstanding technical requirements.

Kind regards,

F. Alex Feltus, Ph.D.

Academic Editor

PLOS ONE

Additional Editor Comments (optional):

Reviewers' comments:

Reviewer's Responses to Questions

**Comments to the Author**

1. If the authors have adequately addressed your comments raised in a previous round of review and you feel that this manuscript is now acceptable for publication, you may indicate that here to bypass the “Comments to the Author” section, enter your conflict of interest statement in the “Confidential to Editor” section, and submit your "Accept" recommendation.

Reviewer #1: (No Response)

Reviewer #2: All comments have been addressed

2. Is the manuscript technically sound, and do the data support the conclusions?

Reviewer #1: Yes

Reviewer #2: Yes

3. Has the statistical analysis been performed appropriately and rigorously? 

Reviewer #1: Yes

Reviewer #2: Yes

4. Have the authors made all data underlying the findings in their manuscript fully available?

Reviewer #1: Yes

Reviewer #2: Yes

5. Is the manuscript presented in an intelligible fashion and written in standard English?

Reviewer #1: Yes

Reviewer #2: Yes

6. Review Comments to the Author

Reviewer #1: (No Response)

Reviewer #2: (No Response)

7. PLOS authors have the option to publish the peer review history of their article (what does this mean?). If published, this will include your full peer review and any attached files.

Reviewer #1: No

Reviewer #2: No

---

## [Editor Report · Acceptance letter]

15 Feb 2021

PONE-D-20-09138R2 

Construction of a reference transcriptome for the analysis of male sterility in sugi (*Cryptomeria japonica* D. Don) focusing on *MALE STERILITY 1 (MS1)***

Dear Dr. Ueno:

I'm pleased to inform you that your manuscript has been deemed suitable for publication in PLOS ONE. Congratulations! Your manuscript is now with our production department. 

Kind regards, 

on behalf of

Dr. F. Alex Feltus 

Academic Editor

PLOS ONE